# High temperature sensitivity is intrinsic to voltage-gated potassium channels

**Fan Yang, Jie Zheng***

Department of Physiology and Membrane Biology, University of California, Davis School of Medicine, Davis, United States

**Abstract** Temperature-sensitive transient receptor potential (TRP) ion channels are members of the large tetrameric cation channels superfamily but are considered to be uniquely sensitive to heat, which has been presumed to be due to the existence of an unidentified temperature-sensing domain. Here we report that the homologous voltage-gated potassium (Kv) channels also exhibit high temperature sensitivity comparable to that of TRPV1, which is detectable under specific conditions when the voltage sensor is functionally decoupled from the activation gate through either intrinsic mechanisms or mutations. Interestingly, mutations could tune Shaker channel to be either heat-activated or heat-deactivated. Therefore, high temperature sensitivity is intrinsic to both TRP and Kv channels. Our findings suggest important physiological roles of heat-induced variation in Kv channel activities. Mechanistically our findings indicate that temperature-sensing TRP channels may not contain a specialized heat-sensor domain; instead, non-obligatory allosteric gating permits the intrinsic heat sensitivity to drive channel activation, allowing temperature-sensitive TRP channels to function as polymodal nociceptors.

## Introduction

Sensitive detection and discrimination of temperature cues are fundamental to the survival and prosperity of humans and animals. While Hodgkin and Huxley showed more than 60 years ago that temperature could profoundly influence membrane excitability (*Hodgkin and Huxley, 1952*), it is in general not well understood how heat affects membrane excitability and how temperature-dependent changes in neuronal activity contribute to physiology. Neuronal action potentials are generated and modulated by a precise combination of ionic currents produced by the activity of ion channels. Activation of ion channels in turn is the result of complex conformational rearrangements in channel protein controlled by specific physical or chemical stimuli (*Hille, 2001*). Heat contributes to activation energy for channel conformational changes, through which it regulates channel activity and the shape of action potential (*Rodriguez et al., 1998*; *Liang et al., 2009*). For most ion channels, thermal energy affects the rate of conformational transitions by 2-to-5 folds per 10°C (*Pusch et al., 1997*; *Rodriguez et al., 1998*). Within the physiological temperature range, thermal energy alone was found to be generally insufficient to initiate channel activation. Among the exceptions are a group of transient receptor potential (TRP) channels including TRPV1-4, TRPM8, TRPM3, TRPM4, TRPM5, TRPC5, and TRPA1. These channels are potently activated by heat in the absence of another stimulus and hence serve as key cellular temperature sensors (*Clapham, 2003*; *Zheng, 2013*). How these TRP channels respond to heat with exquisite sensitivity, however, remains mysterious.

To understand the molecular mechanism underlying high temperature sensitivity of the TRP channels, we conducted a comparative investigation of the voltage-gated potassium (Kv) channels. As members of the tetrameric cation channel superfamily, TRP channels and Kv channels are structurally similar. They all have six transmembrane segments, with the S1 to S4 segments forming an isolated peripheral domain surrounding the ion-permeating pore composed of S5, S6, and the loop between them. Comparison of the crystal structures of Kv channels (*Long et al., 2005*, *2007*) and the cryo-EM structures

*For correspondence: jzheng@ucdavis.edu

**Competing interests:** The authors declare that no competing interests exist.

**Reviewing editor**: Richard Aldrich, The University of Texas at Austin, United States

**eLife digest** If you touch something too hot, it can cause you pain and damage your skin. Sensing the heat given off by an object or the temperature of the environment is possible, at least in part, because of proteins called temperature-sensitive TRP ion channels. These proteins are found in the cell membranes of nerve endings that are underneath the skin; and they open in response to heat, allowing ions to flow into the nerve cell. This in turn triggers a nerve impulse that is sent to our central nervous system and is perceived as heat and/or pain.

The ability to sense heat was thought to be unique to these TRP ion channels, and it was believed that these ion channels contained an as-yet unidentified temperature-sensing domain. However, Yang and Zheng now report that similar ion channels, which open in response to changes in the voltage that exists across a cell's membrane, are also sensitive to changes in temperature.

The temperature response of these 'voltage-gated channels' had largely eluded the attention of researchers in the past. This is because parts of the ion channel—which act like a 'voltage sensor' and only shift when the membrane voltage changes—normally keep the channel closed and directly open the channel when they move. Like all other proteins, ion channels are made from smaller building blocks called amino acids; and by changing some of the amino acids in the voltage-gated channel Yang and Zheng could decouple these normally linked actions. The changes to the channel meant that it did not immediately open when the voltage sensor moved; and decreasing the concentration of calcium ions inside the cell had the same effect as changing these amino acids. Both approaches revealed that, after a change in membrane voltage caused the voltage sensor to move, the ion channel remained closed until a high temperature caused it to open. Yang and Zheng revealed that the response of the modified voltage-gated channel to temperature was comparable to that of a typical heat-sensitive TRP ion channel.

Further experiments showed that replacing some of the amino acids in the voltage-gated potassium ion channel with different amino acids could cause the channel to be either opened or closed by heat.

The findings of Yang and Zheng indicate that temperature-sensing TRP channels may not contain a specialized heat-sensor domain. Instead, as these TRP ion channels do not require other parts of the protein to move in order to open the channel, they can be activated by their own inherent sensitivity to heat.

of TRPV1 (*Cao et al., 2013*; *Liao et al., 2013*) further shows that detailed structural features of these two types of channels are also very similar. However, the two types of channels are functionally distinct. Kv channels are activated by membrane depolarization with a steep voltage dependence (*Sigworth, 1994*), while TRP channels are only weakly activated at highly depolarized voltages (*Zheng, 2013*). More importantly, unlike the temperature-sensitive TRP channels, Kv channels cannot be directly activated by changes in temperature, and the activity of Kv channels are generally considered to be not very heat-sensitive (*Rodriguez et al., 1998*; *Cui et al., 2012*). The differences lead to the widely accepted assumption that through evolution some TRP channels have acquired specific protein structures that serve as a 'heat sensor'. In the present study we tested this hypothesis by examining the temperature response of Kv channels.

## Results

TRPV1 is an archetypical temperature-sensitive TRP channel (*Caterina et al., 1997*). Heat strongly activates the channel, which could be observed at a broad voltage range using a ramp protocol (*Figure 1A*, upper panel). A common way to characterize temperature sensitivity is the $Q_{10}$ value (defined as the folds increase in current amplitude upon a 10°C increase in temperature). For TRPV1, the $Q_{10}$ value is above 20 over a more than 200 mV voltage range (*Figure 1A*, lower panel) (*Benham et al., 2003*), reflecting outstanding sensitivity of channel activation to heat. Having high temperature sensitivity at a wide voltage range is crucial for the channel's physiological role as a temperature sensor—it allows TRPV1-expressing sensory neurons to detect heat no matter the neurons are in the resting state or excited state.

Activity of Shaker potassium channel, in contrast, exhibited much lower temperature sensitivity (*Figure 1B*, upper panel). As a voltage-gated channel, Shaker activates upon depolarization to about

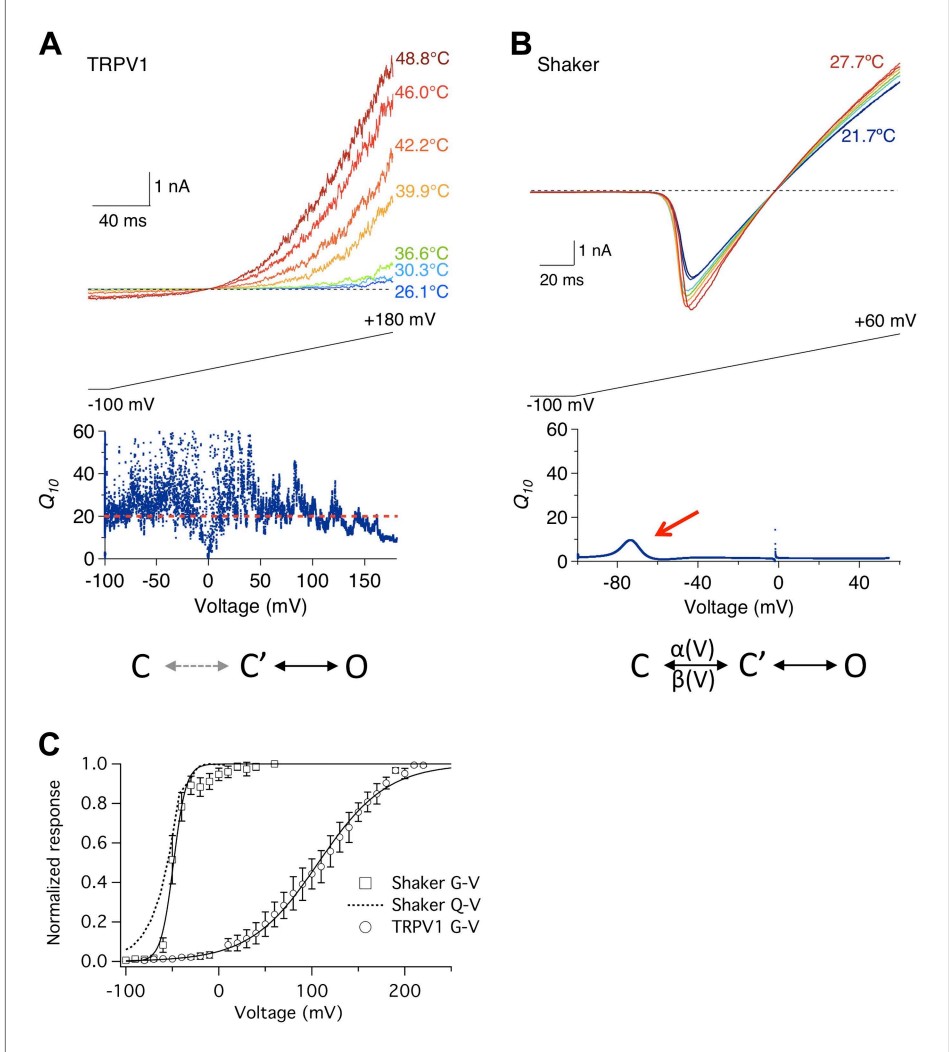

**Figure 1**. Heat sensitivity of TRPV1 and Shaker channel exhibits distinct voltage dependence. (**A**) TRPV1 current was elicited by a voltage ramp at varying temperatures (upper panel). $Q_{10}$, quantified between 36.6°C and 46.0°C, remains high across the entire voltage range from −100 mV to +150 mV (lower panel). (**B**) Shaker channels exhibit a low heat sensitivity at most voltages as quantified by $Q_{10}$ between 22.4°C and 27.7°C. However, $Q_{10}$ transiently peaks (red arrow) around the voltage (−70 mV) where the channel just starts to open. Simplified gating schemes involving two transitions are shown on the bottom. The C→C' transition is weakly voltage-dependent for TRPV1 but highly voltage-dependent for Shaker. (**C**) TRPV1 and Shaker channels show distinct voltage-dependent activation behaviors. TRPV1 conductance (open circle) has a shallow voltage dependence (with an apparent gating charge of 0.76 ± 0.06 $e_0$, $n = 4$) that occurs in a highly depolarized range ($V_{1/2}$ = 114.9 ± 10.9 mV, $n = 4$). Shaker activation has a steep voltage dependence (with an apparent gating charge 5.1 ± 1.3 $e_0$, $n = 5$) that occurs at hyperpolarized voltages ($V_{1/2}$ = −54.2 ± 4.7 mV, $n = 5$). The Q-V curve for Shaker (dotted curve) is reproduced from a published study (***Schoppa and Sigworth, 1998***).

−60 mV (***Figure 1C***). The threshold voltage for activation was only slightly shifted by raising temperature. The average $Q_{10}$ value remained low, at below 4 (***Figure 1B***, lower panel). Low temperature sensitivity is anticipated for Shaker and many other Kv channels, because opening and closing of the ion permeation pore in these channels is obligatorily coupled to movement of the voltage-sensor controlled by the membrane potential (***Sigworth, 1994***). At hyperpolarized voltages, the channel is locked in the initial closed state (C, ***Figure 1B***), in which the voltage-sensor is kept in the down conformation. A strongly voltage-dependent transition, involving the movement of ~13 $e_0$ gating charges across the transmembrane electric field (***Schoppa et al., 1992***; ***Zagotta et al., 1994***;

*Aggarwal and MacKinnon, 1996*), moves the channel to another closed state, C', from which it can transition to the open state, O, with little voltage dependence. Since thermal energy is insufficient to supply the activation energy for voltage-sensor to overcome transmembrane voltage, opening of the channel is dictated by the membrane potential. The high fidelity of Shaker and other voltage-gated channels in reporting changes in membrane potential at variable environmental conditions is the basis for reliable electrical signaling of the nervous system.

A closer inspection of the Shaker channel $Q_{10}$ measurement, however, revealed that it did increase modestly around −80 to −60 mV (arrow in *Figure 1B*, lower panel), approaching 10 at its peak. It is intriguing that this is the voltage range at which the voltage-sensor starts to move, permitting the C→C' transition. This can be seen in the voltage dependence of gating charge movement (Q-V curve, *Figure 1C*). Since it has been previously suggested that the voltage-dependent transition in TRPV1 is highly temperature-sensitive (*Voets et al., 2004*), we wondered whether the transient $Q_{10}$ increase in Shaker might reflect temperature sensitivity of the voltage-sensor movement. To test this possibility, we conducted similar measurements with the voltage-gated $Ca^{2+}$-modulated BK potassium channel, because for this channel the separation between the G-V curve and the Q-V curve can be conveniently controlled by intracellular $Ca^{2+}$.

We observed that, like Shaker, BK in the presence of intracellular $Ca^{2+}$ also exhibited a transient $Q_{10}$ increase at the voltage range where voltage-sensor started to be activated by depolarization, around −80 mV (*Figure 2A*). Removing $Ca^{2+}$ shifts the voltage range for channel activation, substantially increasing the separation between G-V and Q-V curves (*Horrigan and Aldrich, 1999*). The change is achieved mainly through a dual-allosteric coupling between the C'→O transition and the $Ca^{2+}$ and voltage induced transitions (*Horrigan and Aldrich, 1999*). We observed two interesting effects of $Ca^{2+}$ removal on the transient $Q_{10}$ increase. First, it substantially shifted the voltage range at which the transient $Q_{10}$ increase occurred, to around +100 mV (*Figure 2B*). This large shift mirrored the substantial shift of the G-V curve. More importantly, in the absence of $Ca^{2+}$ the extent of the $Q_{10}$ increase was enhanced dramatically even though the total gating charge movement remained unchanged (*Horrigan and Aldrich, 1999*; *Figure 2B*). These observations suggest that the $Q_{10}$ increase may not be associated with the voltage-sensor movement (as reflected by the Q-V curve) but rather associated with the activation gate opening (as reflected by the G-V curve), that is, instead of the C→C' transition, it seemed to be the C'→O transition that yielded the $Q_{10}$ increase.

The possibility of a voltage-sensor associated $Q_{10}$ increase was further ruled out when we tested additional Kv channels. We found that both Kv2.1 and Kv4.3 exhibited a surprisingly prominent $Q_{10}$ transient increase that peaked in the 20-to-30 range (*Figure 3*), a level comparable to that of TRPV1. Highly temperature-sensitive activation could be observed when either a voltage ramp or a voltage step was applied. As these channels have similar voltage dependence to Shaker and BK channels (*Islas and Sigworth, 1999*; *Dougherty et al., 2008*), the substantially higher $Q_{10}$ values cannot come from the voltage-sensor movement. This conclusion is reinforced by the fact that the activity of TRPV1 (and other temperature-sensitive TRP channels) is minimally voltage-sensitive (with less than 1 $e_0$ gating charge; *Figure 1C*) (*Voets et al., 2004*). Indeed, we observed that the energetic effects of heat on the voltage-dependent activation per se are similar for TRPV1 and most Kv channels (*Figure 4A,B*), and the peak $Q_{10}$ values exhibited no correlation with the temperature-dependent shifts in the half-activation voltage (*Figure 4C*).

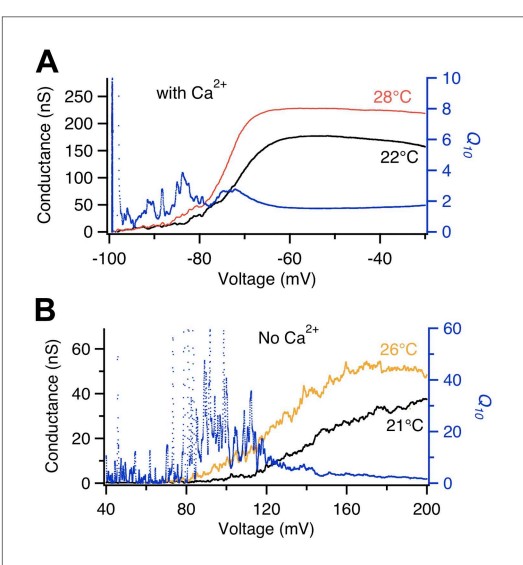

**Figure 2**. Voltage dependence of BK channel temperature sensitivity shifts with intracellular $Ca^{2+}$. $Q_{10}$ (blue trace, right axis) calculated from temperature-dependent changes in the G-V curves at labeled temperatures (left axis) measured in the absence of $Ca^{2+}$ (lower panel) are larger in value and shifted more to the right in voltage dependence compared to those measured in presence of saturating $Ca^{2+}$ (200 μM, upper panel).

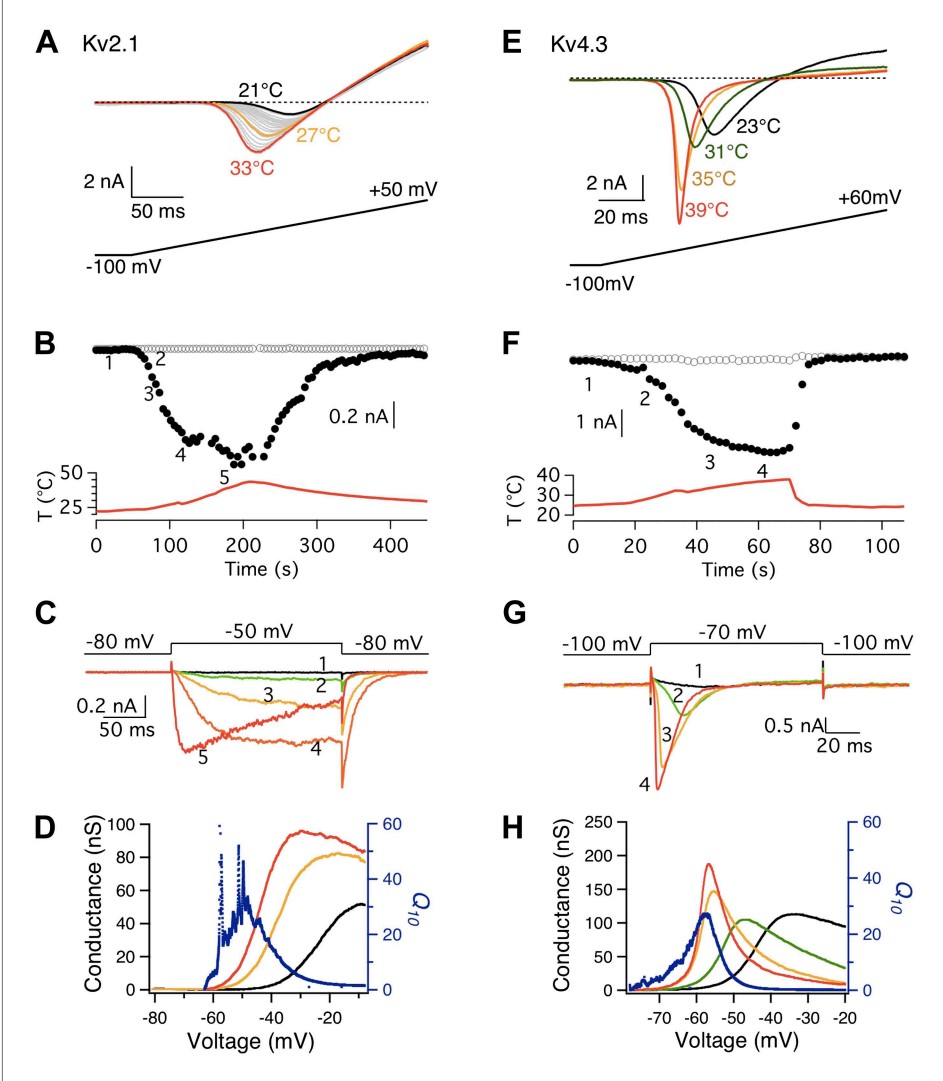

**Figure 3**. Both Kv2.1 and Kv4.3 channels are highly heat-sensitive within a narrow voltage range. (**A**) and (**E**) Large increases in current were observed from both Kv2.1 and Kv4.3 when temperature was raised. (**B**) Kv2.1 channels were substantially activated by heat at −50 mV (filled circles) but not at −80 mV (open circles). (**C**) Representative current traces at time points shown in (**B**). (**F**) and (**G**) Similar behaviors as Kv2.1 were observed with Kv4.3. (**D**) and (**H**) Voltage dependence of G-V curves (left axis) and $Q_{10}$ (right axis) for Kv2.1 (**D**) and Kv4.3 (**H**). G-V curves are color-coded as in (**A**) and (**E**), respectively. $Q_{10}$ peaks in the range where the channel just starts to be voltage-activated.

The following figure supplement is available for figure 3:

**Figure supplement 1**. High heat sensitivity of Kv2.1 channels was robustly observed when the length of voltage ramp and extracellular $K^+$ concentration were changed.

What makes Kv2.1 and Kv4.3 highly temperature-sensitive at a narrow hyperpolarized voltage range? One noticeable common property of these two channels is a prominent closed state inactivation (CSI) process (**Klemic et al., 1998**; **Amadi et al., 2007**; **Figure 5**), during which the voltage-sensor is decoupled from the activation gate (**Klemic et al., 1998**; **Barghaan and Bahring, 2009**). This 'slip-page' in coupling relaxes the obligatory gating by voltage, increasing the probability of transitioning from C into C′ at mild depolarization (**Shin et al., 2004**). In other words, CSI simultaneously makes Kv2.1 and Kv4.3 more TRPV1-like in both voltage- and heat-dependent activation. The $Q_{10}$ increase peaked at roughly −70 to −50 mV, where the probability of the channels being not open but in the CSI state was high (**Figure 5B,C**). Under such condition, both channels could be strongly heat-activated

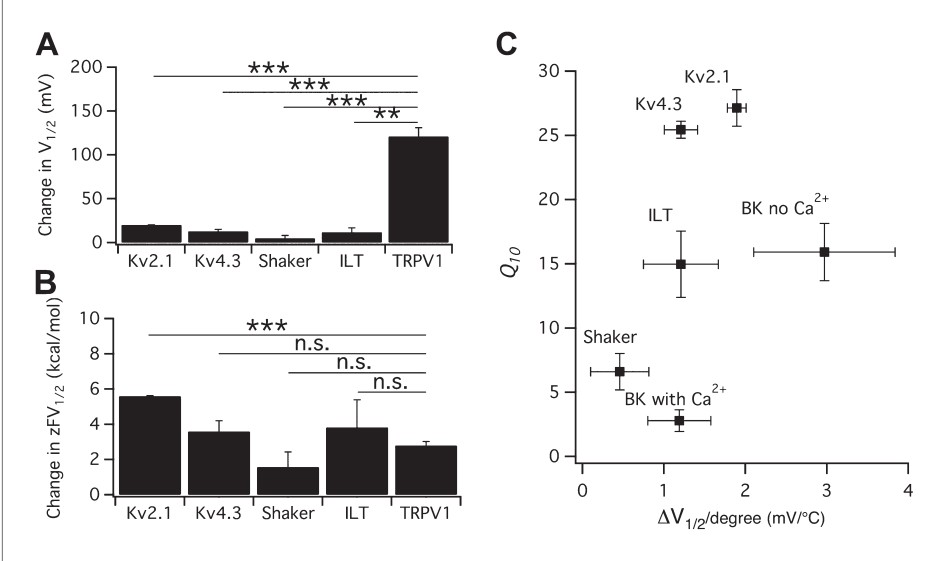

**Figure 4.** Heat-induced shift in $V_{1/2}$ is not responsible for high temperature sensitivity. (**A**) Changes in $V_{1/2}$ were induced by raising temperature from the room temperature to a level that maximally activated the channels ($n$ = 3-to-4). (**B**) Corresponding changes in free energy were calculated as $z \cdot F \cdot \Delta V_{1/2}$, where $z$ is the total gating charge, $F$ is Faraday's constant. $n$ = 3-to-4. Values of $z$ for Kv channels are based on published work (**Schoppa and Sigworth, 1998**; **Islas and Sigworth, 1999**; **Dougherty and Covarrubias, 2006**). \*\*p<0.01; \*\*\*p<0.001; n.s., no significance. (**C**) $Q_{10}$ value of each channel type was plotted against the temperature-induced shift in $V_{1/2}$. At similar levels of shift in $V_{1/2}$, both large and small $Q_{10}$ values were observed, suggesting that shift in $V_{1/2}$ is not responsible for high temperature sensitivity. $n$ = 3-to-4.

(**Figure 3**). In contrast to Kv2.1 and Kv4.3, the less temperature-sensitive Shaker channel does not undergo CSI (**Aldrich, 2001**; **Zhou et al., 2001**; **Fineberg et al., 2012**), so that its voltage-sensor is always obligatorily coupled to the activation gate, preventing the gate being activated by heat. To rule out the possibility that transient $Q_{10}$ increase was produced by artificial non-equilibrium gating generated by the voltage ramp protocol, we repeated the experiment with Kv2.1 using a much slower ramp protocol (from 200 ms to 1 s duration) (**Figure 3—figure supplement 1A,B**). We also replaced extracellular potassium with sodium in order to observe channel activation as an outward current instead of an inward current (**Figure 3—figure supplement 1C,D**). Neither operation eliminated the large transient $Q_{10}$ increase.

If the $Q_{10}$ increase in Kv2.1 and Kv4.3 is due to relaxation of the obligatory coupling between the voltage-sensor and the activation gate through CSI, one would expect to be able to produce a larger $Q_{10}$ value in Shaker by decoupling its voltage-sensor from the activation gate. Two well-characterized Shaker mutants are known to exhibit this property, ILT (**Smith-Maxwell et al., 1998**) and V2 (**Schoppa and Sigworth, 1998**). The mutations in ILT generate a prominent separation between the Q-V curve and the G-V curve (**Figure 6A**). When we examined the ILT channel, we indeed observed a large $Q_{10}$ transient that peaked above 20, making activation of this channel much more temperature-sensitive than the wild-type channel (**Figure 6B–E**). Importantly, the $Q_{10}$ transient started near −20 mV where the voltage-sensor has been mostly elevated by depolarization. Thus, the $Q_{10}$ transient seen in ILT and other Kv channels is not originated from the voltage-sensor movement. On the contrary, our observations argue strongly that it is the release of voltage-sensor control—through depolarization, CSI, and mutations—that revealed an intrinsically highly temperature-sensitive process in Kv channels that occurs during the C′→O transition.

The Shaker V2 mutant also has a large separation between Q-V and G-V curves (**Figure 6F**). Consistent with the conclusions we postulate above, V2 also exhibited a noticeable $Q_{10}$ change at the voltage range for channel opening. Interestingly, we observed that instead of heat activation, the channel exhibited a heat-induced deactivation behavior (**Figure 6G–J**). At voltages where the voltage-sensor has been fully elevated, V2 current decreased substantially upon heating. This yielded $Q_{10}$ values

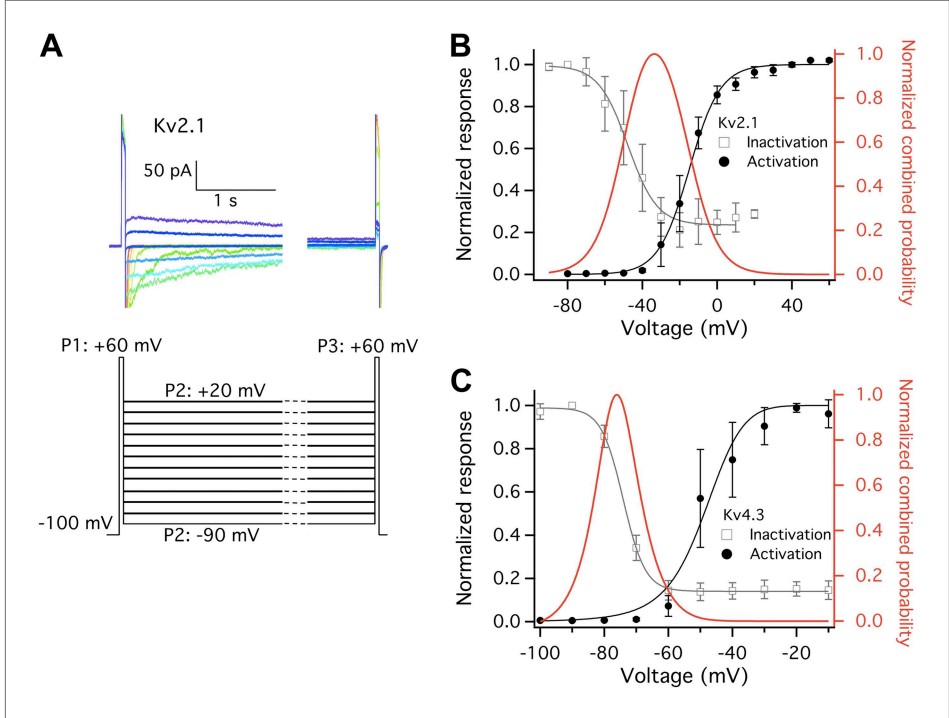

**Figure 5**. Closed state inactivation (CSI) and channel activation overlap within specific voltage ranges. (**A**) Representative current recordings of Kv2.1 CSI. Length of the P2 segment was 10 s. Current measured at P3 was normalized to that measured at P1. The normalized current was then plotted against P2 voltage for voltage dependence of CSI. (**B**) For Kv2.1, the voltage dependence of both CSI (open squares) and activation (filled circles) follows a single-Boltzmann function (left axis). After normalization the two Boltzmann functions were multiplied to yield the red curve (right axis). This combined probability curve surges in the voltage range where the channels are only slightly activated while significant CSI occurs, which overlaps with the voltage range where high heat sensitivity is observed. (**C**) Similar voltage dependences of CSI and activation were observed for Kv4.3. n = 3-to-5.

less than 1, as seen in the cold-activated TRPM8 channel (**McKemy et al., 2002**). It remains to be determined whether the heat-induced deactivation process in V2 reflects a reversal of the heat-induced activation process in wild-type Kv channels and ILT. It is nonetheless important to note that V2 and ILT differ by only a few amino acids (V2 contains L382V in the S4–S5 linker; ILT contains V369I, I372L, and S376T in the S4 segment). Given that these mutations appear to alter gating transition rates and equilibriums without changing the fundamental voltage-dependent activation mechanism (**Schoppa and Sigworth, 1998**; **Smith-Maxwell et al., 1998**), we speculate that the same gating process may yield the opposite $Q_{10}$ changes. While future investigation is needed to examine this particular speculation and identify the gating transition involved, it is important to note that V2 is also intrinsically heat-sensitive, which is revealed by decoupling the voltage-sensor from the activation gate.

In summary, we found that gating of Kv channels can be highly temperature-sensitive (**Figure 7**). Potentially this transient increase in temperature sensitivity may have important functional consequences, especially since a change in the Kv channel activity at slightly depolarized membrane potentials would substantially alter action potential frequency and neuronal coding. Similarly, any temperature-dependent change in activity of the voltage-gated sodium and calcium channels may also have important consequence in membrane excitability. However, while it has been demonstrated that temperature-dependent changes in Kv channel activity profoundly affect neuronal excitability and coding (**Vandenberg et al., 2006**; **Graham et al., 2008**) as well as secretion (**MacDonald et al., 2003**), how dynamic changes in temperature sensitivity of ion channels affect physiology remains poorly understood, partially because most studies of ion channel behaviors have been carried out at the room temperature instead of around the body temperature.

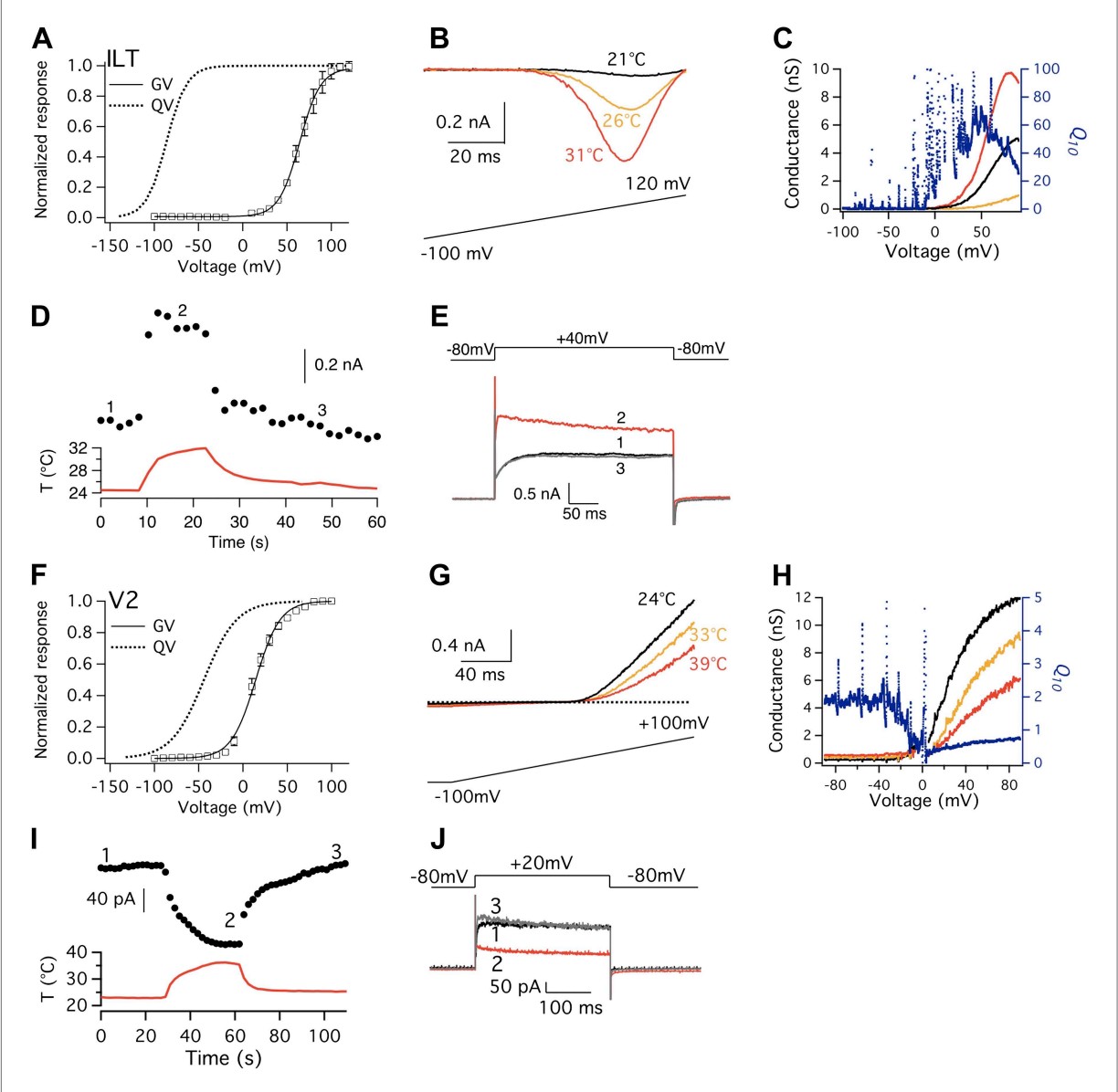

**Figure 6**. Shaker ILT and V2 mutants exhibit large but opposite heat responses. (**A**) and (**F**) Conductance-voltage (G-V; *n* = 3-to-6) and gating charge-voltage (Q-V) curves for ILT (**A**) and V2 (**F**). Q-V curves for ILT and V2 are reproduced from published studies (*Schoppa and Sigworth, 1998*; *Ledwell and Aldrich, 1999*). (**B**) Large increases in current are observed from ILT when temperature was raised. To increase current amplitude when driving voltage was small, extracellular solution contained 130 mM KCl, while intracellular solution contained 130 mM NaCl. (**C**) Voltage dependence of $Q_{10}$ for ILT based on G-V curves derived from (**B**). (**D**) and (**E**) ILT channels were substantially activated by heat at +40 mV. (**G**) In contrast to the ILT current, V2 current decreases upon temperature rise. (**H**) Voltage dependence of $Q_{10}$ for V2 based on G-V curves derived from (**G**). $Q_{10}$ drops below 1 when voltage reaches a level that the channel starts to open, as increasing temperature deactivates the channel. (**I**) and (**J**) V2 channels were substantially deactivated by heat at +20 mV.

## Discussion

A central observation of the present study is that there is a transient increase in temperature sensitivity of Kv channels, which occurs when the voltage-sensor is partially decoupled from the activation gate. Our study revealed an intrinsically heat-sensitive gating process in these voltage-gated channels. Interestingly, a previous study of Shaker channel activation indeed identified a late transition preceding channel opening that exhibits higher temperature sensitivity than all the other transitions (including the voltage-sensor movements) (*Rodriguez et al., 1998*). In agreement of *Clapham and Miller (2011)*,

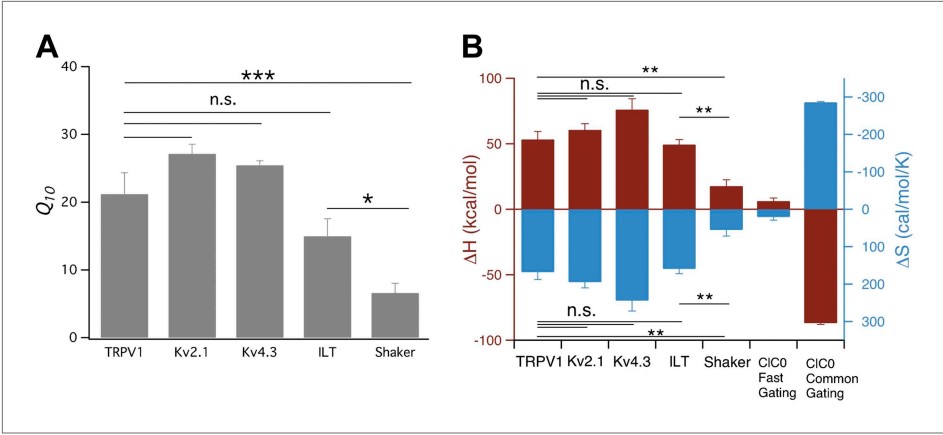

**Figure 7**. Comparison of $Q_{10}$ (**A**) and thermodynamic measurements (**B**) between TRPV1, Kv and CLC-0 channels. Data for CLC-0 are reproduced from published results (**Yang et al., 2010**). n = 3-to-6; *p<0.05; **p<0.01; ***p<0.001; n.s., no significance.

these observations indicate that there may not be a specialized 'heat sensor' to support the heat activation process. Generally speaking, an equilibrium process A←→B would be highly temperature-sensitive if the transition is associated with a large enthalpic change. This can be seen from the temperature dependence of the probability function

$$P_B = \frac{1}{1 + \exp\left(\dfrac{\Delta H}{RT} - \dfrac{\Delta S}{R}\right)},$$

in which $P_B$ is the probability of the system to reside in the B state, $\Delta H$ and $\Delta S$ are enthalpic and entropic changes associated with the transition, respectively, T is temperature, and R is the gas constant. For temperature-sensitive TRP channels, a large enthalpic change during channel activation was indeed observed (**Brauchi et al., 2004**; **Voets et al., 2004**; **Yang et al., 2010**). The observation of an associated large entropic change comes from the requirement for the heat-induced transition to occur at physiological temperatures (**Yang et al., 2010**). A simple interpretation of the large enthalpic and entropic changes is that they represent the occurrence of a substantial conformational rearrangement in the channel protein. We showed in the present study that in Kv channels this conformational rearrangement is not the voltage-sensor movement but rather the process following voltage-sensor movement.

Kv channels and TRP channels are known to function as an allosteric protein. Observations from the present study can be understood using the following simple allosteric gating scheme:

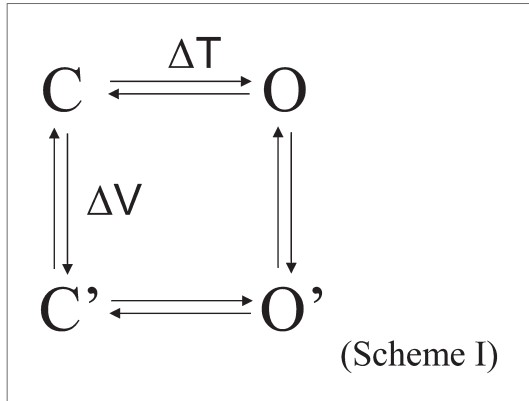

in which the vertical transition represents voltage-sensor movement driven by membrane depolarization, and the horizontal transition represents pore opening. Because for Shaker and many other Kv channels voltage-sensor movement is very strongly coupled to pore opening, activation gating occurs in the sequence C↔C'↔O'. At hyperpolarized voltages, the strong voltage dependence of the C↔C' equilibrium ensures that the channel resides in the initial C state, so that the temperature-sensitive transition C'→O' cannot occur. (According to allosteric principles, the C'→O' transition is strongly favored over the C→O transition because of the large activation energy associated with the vertical transition.) Only when the channel moves from C to C' upon mild depolarization can the C'→O' transition be observed upon temperature changes. CSI in Kv2.1 and Kv4.3 as well as mutations in Shaker ILT and V2 weaken the allosteric coupling between the voltage-sensor and the pore, allowing the temperature-sensitive horizontal transition to occur.

Realizing that Kv channels are intrinsically highly temperature-sensitive has important mechanistic implications for how some TRP channels serve as temperature sensors and nociceptors. The high temperature sensitivity of TRP channels originates from exceptionally large enthalpic and entropic changes associated with the activation process (*Brauchi et al., 2004*; *Voets et al., 2004*; *Yang et al., 2010*). From the Gibbs equation, $\Delta G = \Delta H - T\Delta S$, it is clear that large $\Delta H$ and $\Delta S$ values make free energy change associated with the activation process, $\Delta G$, to be highly temperature-dependent. It is generally assumed that the large enthalpic and entropic changes come from a substantial conformational rearrangement in the TRP channels. The existence of large conformational rearrangements has been directly detected using fluorescent tags in TRPV1 as well as the CLC chloride channels (whose common gating process is also highly temperature-sensitive) (*Pusch et al., 1997*; *Bykova et al., 2006*; *Yang et al., 2010*; *Ma et al., 2011*). Nonetheless, the structural and mechanistic basis for large conformational rearrangements in temperature-sensitive TRP channels remains unknown. Consequently, it has been highly controversial how these TRP channels sense heat. Here we show that in Kv channels the voltage-dependent C→C' transition, representing movements of the voltage-sensor, is not specially temperature sensitive, in agreement with a previous study of the Shaker channel (*Rodriguez et al., 1998*). Instead, the C'→O' transition, representing later activation transitions of the channel protein, yields the transient increase in $Q_{10}$ when the obligatory voltage-sensor control is alleviated. We and others showed that temperature-sensitive TRP channels also behave like an allosteric protein (*Latorre et al., 2007*; *Matta and Ahern, 2007*; *Jara-Oseguera and Islas, 2013*; *Yang et al., 2014*). High temperature sensitivity of TRP channels may also originate from the C'→O' transition. A previous study demonstrated, based on a simple two-state C↔O gating scheme, that heat strongly shifts the gating equilibrium by deferentially affecting the forward and backward transitions which are also weakly voltage-sensitive (*Voets et al., 2004*). The origin of the voltage-sensitive process in TRPV1 and other TRP channels remains unclear. It is nonetheless becoming clear that an S4-type voltage-sensor seen in Kv channels does not exist in TRP channels. Since voltage is a poor activator for TRPV1 but heat is a strong activator, it is unlikely heat solely works through the voltage-dependent process to open TRPV1 (*Matta and Ahern, 2007*; *Yang et al., 2010*). In the context of Scheme I, it appears that the observation by *Voets et al. (2004)* is applicable to the horizontal transition instead of the vertical transition. Our study indicates that TRPV1 is heat-sensitive at a broad voltage range because the C→C' transition is weakly voltage-dependent and not obligatory.

A recent cryo-EM structural study of TRPV1 provided direct evidence that the peripheral S1–S4 domains remain stationary while the pore domains undergo substantial structural rearrangements when the channel is activated by agonists (*Cao et al., 2013*; *Liao et al., 2013*). It is thus possible that the highly temperature-sensitive process of TRPV1 may come from the pore opening process. Indeed, nearly all the known TRPV1 agonists interact directly or indirectly with the pore domain: $H^+$ (*Jordt et al., 2000*), spider toxin (*Bohlen et al., 2010*), and divalent cations (*Ahern et al., 2005*; *Luebbert et al., 2010*; *Cao et al., 2014*; *Yang et al., 2014*) from the extracellular side, and capsaicin and resiniferatoxin (*Szallasi et al., 1999*; *Jordt and Julius, 2002*) from the intracellular side. While capsaicin and resiniferatoxin bind to sites adjacent to the pore-forming domains, the cryo-EM data indicate that they induce channel activation by altering the pore conformation while leaving the S1–S4 peripheral domains mostly unchanged (*Cao et al., 2013*; *Liao et al., 2013*). Results from previous studies on mutant channels (*Myers et al., 2008*; *Grandl et al., 2010*; *Cui et al., 2012*) and site-directed fluorescence recordings (*Yang et al., 2010, 2014*) further support the involvement of the outer pore in heat activation. Can conformational rearrangement of the channel pore

produce the large $\Delta H$ and $\Delta S$ required for high temperature sensitivity? Results from the cryo-EM study (*Cao et al., 2013*) and fluorescence study (*Yang et al., 2010*, *2014*) of TRPV1 clearly support this possibility.

To serve as biological heat sensors, temperature-sensitive TRP channels need to fulfill two fundamental requirements. First, its activity must be highly sensitive to changes in temperature. Second, heat-induced activity cannot be dependent on the presence of another stimulus; specifically, heat activation must be able to occur when the sensory neuron is in the resting state (at hyperpolarized membrane potentials). As discussed above, the first requirement is fulfilled because channel activation is associated with large $\Delta H$ and $\Delta S$ originated from a substantial conformational rearrangement. Results from the present study argue that the second requirement is fulfilled because the occurrence of heat-induced conformational rearrangements is not controlled by the voltage-dependent transition (or other stimulus-induced transitions). Allosteric but non-obligatory coupling of multiple stimuli to the opening of channel pore allows TRP channels to serve as polymodal cellular sensors.

## Materials and methods

### cDNA constructs and cell transfection

The following cDNA constructs were used in this study: murine TRPV1 (a gift from Dr Michael X Zhu, University of Texas Health Science Center at Houston), Kv2.1, Shaker IR channel (with N terminal inactivation ball removed, used as WT) and Shaker IR L382V (V2 mutant) (all gifts from Dr Jon Sack, University of California at Davis), Shaker IR-based V369I, I372L, and S376T triple mutant (ILT mutant) (a gift from Dr Kenton Swartz, National Institutes of Health at Bethesda), Kv4.3 (a gift from Dr Kewei Wang, Peking University in China), and murine Slo1 construct for BK channel pore-forming subunit (a gift from Dr Jim Trimmer, University of California at Davis). To facilitate identification of channel-expressing cells, fluorescence protein eYFP was fused in frame to the C-terminus of TRPV1, while GFP was attached to the C-terminus of Kv4.3 and mSlo1 and N-terminus of Kv2.1. Fusion of these fluorescence proteins did not change channel function, as previously described (*Antonucci et al., 2001*; *Giraldez et al., 2005*; *Cheng et al., 2007*; *Cui et al., 2008*).

HEK293 cells were cultured at 37°C with 5% $CO_2$ in a DMEM medium with 10% FBS, 100 U/ml penicillin and 100 mg/ml streptomycin. Cells were transiently transfected with ~0.8 μg cDNA using Lipofectamine 2000 according to the manufacturer's instruction (Invitrogen, Grand Island, NY). Electrophysiogical experiments were performed 24 to 48 hr after transfection.

### Electrophysiological recordings

Macroscopic currents were recorded in the inside-out configuration using a HEKA EPC10 amplifier with PatchMaster software (HEKA, Germany). Patch pipettes were pulled from borosilicate glass and fire-polished to a resistance of ~2 MΩ. The conductance-voltage (G-V) curve was determined from currents in response to a series of voltage steps starting from a deep hyperpolarized voltage (in most cases −100 mV).

To record heat responses, both voltage-step and voltage-ramp protocols were applied. For voltage-step protocols, membrane potential was first clamped at a deep hyperpolarized voltage to fully close all channels. For Kv2.1, Kv4.3 and Shaker ILT channels, it was then stepped to a depolarized voltage that activated the channels to 1-to-3% of their maximum voltage activation according to the G-V curves determined previously. For Shaker V2, transmembrane voltage was stepped to a level that generated ~60% maximum response, as heating would deactivate the channel. The transmembrane voltage was then stepped back to the original hyperpolarized level to close the channels. For voltage-ramp protocols, membrane potential was again first clamped at a hyperpolarized voltage, from which the voltage was linearly ramped up to a depolarized voltage that maximally activated the channels according to the G-V curves. Data were filtered at 2.8 kHz and sampled at 10.0 kHz.

For TRPV1 channel, both bath solution and pipette solution contained 130 mM NaCl, 0.2 mM EDTA and 3 mM Hepes (pH 7.2). For potassium channels and their mutants, both bath solution and pipette solution contained 130 mM KCl, 0.2 mM EDTA and 3 mM Hepes (pH 7.2) unless otherwise stated.

### Temperature control and monitoring

To heat the membrane patch containing ion channels, the bath solution heated by an SHM-828 eight-line heater driven by a CL-100 temperature controller (Harvard Apparatus, Holliston, MA) was perfused

to the pipette tip. A custom-made manifold was attached to the output ports of the heater to increase flow volume and to provide heat insulation. To accurately monitor local temperature at the pipette tip, we placed the IT-24P ultrafine thermocouple bead of a BAT-12 microprobe thermometer (Physitemp, Clifton, NJ) less than 1 mm from the pipette tip. HEKA patch-clamp amplifier registered temperature readings from the thermometer simultaneously with current recording. The speed of temperature change was set at a moderate rate of about 0.3°C/s. This rate ensured that heat-driven gating transitions of the channels reached equilibrium during the course of temperature change, so that the channels were recorded at the equilibrium state. When the experimental temperature was not controlled, recordings were conducted at room temperature at 24°C.

## Data analysis: leak subtraction

Leak current was subtracted from raw current recordings before any data analysis was performed. Leak conductance was determined as the current measured at −100 mV divided by the driving voltage. The level of leak current at each testing voltage was calculated as the product of this leak conductance and the driving voltage.

## G-V curves

G-V curves were fitted to a single-Boltzmann function:

$$\frac{G}{G_{max}} = \frac{1}{1 + e^{-\frac{qF}{RT}(V - V_{1/2})}},$$ (1)

where G/Gmax is the normalized conductance, $V_{1/2}$ is the half-activation voltage, q is the apparent gating charge and F is Faraday's constant.

## Thermodynamic analysis

Heat sensitivity of each channel type was quantified by the enthalpic change, $\Delta H$, and the entropic change, $\Delta S$, associated with the heat-induced activation process. To measure these thermodynamic parameters, we constructed a Van't Hoff plot from highly temperature-sensitive current changes and fitted it to the following equation:

$$\ln K_{eq} = -\frac{\Delta H}{RT} + \frac{\Delta S}{R},$$ (2)

where $K_{eq}$ is the equilibrium constant for heat-driven activation calculated from the channel open probability, R is the gas constant, and T is the temperature in Kelvin. Current was measured at a voltage that induced 1-to-3% channel open probability according to its G-V curve. The $Q_{10}$ value of temperature-dependent single-channel conductance increase was set at 1.5, with which the macroscopic current amplitude at different temperatures was corrected as if the temperature were 22°C (*Yang et al., 2010*). For Kv channels, the conductance-corrected G-V curves were normalized to published maximum open probability. For TRPV1, corrected-conductance was normalized to the conductance elicited by saturating concentration (3 µM) of capsaicin. Open probability determined this way was further used to calculate $K_{eq}$.

## $Q_{10}$ calculation

Another way to quantify heat response was $Q_{10}$ measurement. We first measured current amplitude ($I_1$) at the threshold temperature ($T_1$) where heat activation started. The current amplitude ($I_2$) at a temperature approximately 10°C higher ($T_2$) was measured for TRPV1, Kv2.1, Kv4.3, Shaker ILT and Shaker V2 channels. For Shaker WT channel $I_2$ was measured at a temperature 5°C higher than $T_1$ because further elevation of the experimental temperature led to channel inactivation. $Q_{10}$ was calculated as:

$$Q_{10} = \left(\frac{I_2}{I_1}\right)^{\frac{10}{T_2 - T_1}}.$$ (3)

## Statistics

All statistical data are given as mean ± SEM. Student's *t* test was used to examine the significance of statistical differences: *$p < 0.05$; **$p < 0.01$; ***$p < 0.001$; n.s., no significance.

## Acknowledgements

We are grateful to Drs Toshi Hoshi, Jon Sack, Kenton Swartz, James Trimmer, KeWei Wang, Gary Yellen, and Michael X Zhu for sharing cDNA constructs, Dr Yiquan Tang for stimulating discussion, and our lab members for assistance. Research in the Zheng lab is supported by a National Institutes of Health grant R01NS072377 (to JZ). FY was a recipient of the American Heart Association predoctoral fellowship (10PRE4170142) when this work was carried out.

## Additional information

### Funding

| Funder | Grant reference number | Author |
| --- | --- | --- |
| National Institute of Neurological Disorders and Stroke | R01NS072377 | Jie Zheng |
| American Heart Association | 10PRE4170142 | Fan Yang |

The funders had no role in study design, data collection and interpretation, or the decision to submit the work for publication.

### Author contributions

FY, Conception and design, Acquisition of data, Analysis and interpretation of data, Drafting or revising the article; JZ, Conception and design, Analysis and interpretation of data, Drafting or revising the article

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
