## [Decision Letter]

Thank you for sending your work entitled “High Temperature Sensitivity Is Intrinsic to Voltage-Gated Potassium Channels” for consideration at *eLife*. Your article has been favorably evaluated by Eve Marder (Senior editor) and 3 reviewers, one of whom is a member of our Board of Reviewing Editors.

The following individuals responsible for the peer review of your submission have agreed to reveal their identity: Richard Aldrich (Reviewing editor); David Clapham, and Haoxing Xu (peer reviewers).

The Reviewing editor and the other reviewers discussed their comments before we reached this decision, and the Reviewing editor has assembled the following comments to help you prepare a revised submission.

Yang and Zheng present very interesting results on the temperature dependence of potassium channel activation. They find that several different potassium channels have the high temperature dependence characteristic of thermosensitive TRP channels, but only in certain voltage ranges where, the authors argue, activation is partially “decoupled” to voltage sensors. The results are convincing and the similar behavior of several wild type and mutant channels gives considerable strength to the general principles. The overall conclusion is that high temperature sensitivity is not unique to TRP channels and in Kv channels becomes apparent when focusing on the activation transition (reflected in G-V) rather than the voltage sensor movement (Q-V).

The recordings are of high quality and the results are carefully interpreted. Overall, this is a study of general interest, especially to the investigators working on ion channels and sensory transduction.

However, it is important for the authors to establish the specificity of their Q10 measurement and calculation.

1) Additional control experiments are necessary to show that the voltage-dependence of Q10 change is not sensitive to the experimental conditions, such as the rates of the voltage ramp. This is important because the voltage ramps are not ideal for studying voltage-gated channels, and it appears that the kinetics of channel activation and fast inactivation are also highly temperature-sensitive (see Figure 3). For Kv4.3, the outward current is heat-deactivated while the inward current is heat-activated (see Figure 3)! Note that the heat-deactivation of V2 mutant channels was only shown with the outward current. In this regard, the authors may wish to verify their major conclusion by investigating the temperature sensitivity of the outward currents at depolarizing voltages above Ek using physiological concentrations of extracellular K+ (i.e., 3-5 mM). Note that extracellular K+ would also have profound effects on C-type inactivation, which could be temperature-sensitive. We anticipate that these additional control experiments should be easily accomplished.

2) The authors' explanation in terms of uncoupling makes sense, but they stop short of a more sophisticated, and perhaps clearer, mechanistic explanation. Their allosteric explanation could be more precisely formulated with a general state diagram. They should be able to estimate the relative energetic contributions of heat and voltage sensor charge movement to opening the channel. One way of thinking about the results is that heat can't affect open probability under conditions where it would have to “overcome” the resting state influence of the voltage sensors, effectively having to “drag” the voltage sensors to the active state. But at intermediate voltages some of the voltage sensors are already activated so that the energy supplied by heat can overcome the remaining resting voltage sensors. Similar thinking could be applied to the closed-state inactivation and to the voltage-sensor mutant channels.

3) As you show here, perhaps it best to excise the term “ThermoTRP”. Several channels besides TRPs are known to have high Q10 (e.g. Hv1). Although initially used as advertisement, perhaps it is time we retire this phrase. It has never had any useful meaning given that only a few of the 28 TRPs have been studied in any detail, and those because they bind plant agonists that induce sensations of cold or heat.

4) We suggest you rewrite the Introduction with more precision and regard for the sophistication of likely readers. For example, the phrase 'temperature sensing domain” implies that there is a domain, like the voltage sensor domain, that moves in response to temperature changes rather than to a voltage jump. This is not an analogous situation. Best is to introduce the topic as one of protein conformational changes with temperature in general, that there is relatively little examination of the temperature dependence of gating in most of the >300 channels, and then introduce the recent strong interest in temperature dependence in TRP channels. Don't oversimplify by saying that there are known domains that 'sense' delta T. You finally hit the right theme when you say “our findings strongly indicate that the temperature-sensing TRP channels may not contain a specialized heat-sensor domain; instead, non-obligatory allosteric gating permits the intrinsic heat sensitivity to drive channel activation.” This was pointed out by Clapham and Miller, PNAS, and it would be appropriate to reference it.

5) You should refrain from stating “it is prohibited in Kv channels by the tight coupling of activation to the voltage-controlled gating process”. You can say that this is inferred. If you do want to state this, you will have to say how tight coupling of activation to voltage affects temperature sensitivity of gating in some mechanistic way, which means knowing the energies of interactions in detail.

6) The authors may wish to comment on the strong voltage-dependence of Q10 reported for thermoTRPs by Voets et al (Nature 2004).

---

## [Author Response]

*The recordings are of high quality and the results are carefully interpreted. Overall, this is a study of general interest, especially to the investigators working on ion channels and sensory transduction. However, it is important for the authors to establish the specificity of their Q10 measurement and calculation*.

Details of Q10 measurement and calculation are given in the Materials and methods section. Specific temperature ranges for Q10 calculation are given in figure legends.

1) Additional control experiments are necessary to show that the voltage-dependence of Q10 change is not sensitive to the experimental conditions, such as the rates of the voltage ramp. This is important because the voltage ramps are not ideal for studying voltage-gated channels, and it appears that the kinetics of channel activation and fast inactivation are also highly temperature-sensitive (see Figure 3). For Kv4.3, the outward current is heat-deactivated while the inward current is heat-activated (see Figure 3)! Note that the heat-deactivation of V2 mutant channels was only shown with the outward current. In this regard, the authors may wish to verify their major conclusion by investigating the temperature sensitivity of the outward currents at depolarizing voltages above Ek using physiological concentrations of extracellular K+ (i.e., 3-5 mM). Note that extracellular K+ would also have profound effects on C-type inactivation, which could be temperature-sensitive. We anticipate that these additional control experiments should be easily accomplished.

Our conclusions were based on observations from both voltage ramps and voltage steps experiments. With voltage steps experiments channel gating achieves equilibrium. At such equilibrium state significant heat activation was observed for Kv2.1 (Figure 3), Kv4.3 (Figure 3) and Shaker ITL (Figure 6). The ramp protocol indeed does not guarantee equilibrium gating, but it is very effective in illustrating the transient nature of Q10 changes over a wide voltage range. In response to the critique, we have now conducted the ramp experiments at a substantially slower ramping speed. We also repeated the experiments under condition that yielded an outward current, as suggested. Neither operation eliminated the large transient Q10 increase. These results are now included in the revised manuscript as Figure 3—figure supplement 1.

2) The authors' explanation in terms of uncoupling makes sense, but they stop short of a more sophisticated, and perhaps clearer, mechanistic explanation. Their allosteric explanation could be more precisely formulated with a general state diagram. They should be able to estimate the relative energetic contributions of heat and voltage sensor charge movement to opening the channel. One way of thinking about the results is that heat can't affect open probability under conditions where it would have to “overcome” the resting state influence of the voltage sensors, effectively having to “drag” the voltage sensors to the active state. But at intermediate voltages some of the voltage sensors are already activated so that the energy supplied by heat can overcome the remaining resting voltage sensors. Similar thinking could be applied to the closed-state inactivation and to the voltage-sensor mutant channels.

We indeed think along the same line but apparently failed to articulate it clearly in the original manuscript. A state diagram is now used to illustrate different gating behaviors originated from the same allosteric coupling mechanism. Please see sections in Discussion highlighted in yellow.

3) As you show here, perhaps it best to excise the term “ThermoTRP”. Several channels besides TRPs are known to have high Q10 (e.g. Hv1). Although initially used as advertisement, perhaps it is time we retire this phrase. It has never had any useful meaning given that only a few of the 28 TRPs have been studied in any detail, and those because they bind plant agonists that induce sensations of cold or heat.

We have removed this term from the manuscript. Indeed, only a few TRP and non-TRP channels have been carefully examined for heat-induced activation. Observations of the present study suggest that perhaps heat-induced channel activation is more common than anticipated. It is likely not obvious in channels specialized in detecting other particular stimuli, since in such cases allosteric coupling between the stimulus-driven process and channel opening is expected to be strong. The Kv channels we examined in the present study serve as examples (at various degrees) of this type of specialized channels.

4) We suggest you rewrite the Introduction with more precision and regard for the sophistication of likely readers. For example, the phrase 'temperature sensing domain” implies that there is a domain, like the voltage sensor domain, that moves in response to temperature changes rather than to a voltage jump. This is not an analogous situation. Best is to introduce the topic as one of protein conformational changes with temperature in general, that there is relatively little examination of the temperature dependence of gating in most of the >300 channels, and then introduce the recent strong interest in temperature dependence in TRP channels. Don't oversimplify by saying that there are known domains that 'sense' delta T. You finally hit the right theme when you say “our findings strongly indicate that the temperature-sensing TRP channels may not contain a specialized heat-sensor domain; instead, non-obligatory allosteric gating permits the intrinsic heat sensitivity to drive channel activation.” This was pointed out by Clapham and Miller, PNAS, and it would be appropriate to reference it.

We have rewritten the Introduction section as suggested. Our observations are fully consistent with the idea discussed in the Clapham and Miller article, that it is possible TRP channels operate in the absence of a contained heat sensor. The article is now cited in the revised manuscript; thanks for pointing out this oversight in the first submission.

5) You should refrain from stating “it is prohibited in Kv channels by the tight coupling of activation to the voltage-controlled gating process”. You can say that this is inferred. If you do want to state this, you will have to say how tight coupling of activation to voltage affects temperature sensitivity of gating in some mechanistic way, which means knowing the energies of interactions in detail.

We have removed the statement from the Abstract. While the coupling between the voltage sensor and the gate is intensively studied in Kv channels, that of TRPV1 is much less clear. Existing studies (including our recently published study, JGP 2014) demonstrated that allosteric coupling between the voltage-dependent transition and the closed-to-open transition of the pore can be used to satisfactorily describe voltage-dependent gating of TRPV1. In particular, the coupling between voltage and channel activation is much weaker in TRPV1 than that in most Kv channels. This is evident in the low open probability TRPV1 can reach upon strong depolarization, as well as the fact that the channel can be activated by ligand and heat at hyperpolarized voltages. This difference and its potential implication in the activation mechanism are discussed in the revised manuscript in a speculative manner.

6) The authors may wish to comment on the strong voltage-dependence of Q10 reported for thermoTRPs by Voets et al (Nature 2004).

In the revised manuscript we discussed this previous study and pointed out that the origin of voltage sensitivity in temperature-sensitive TRP channels remains unknown.